# EFFICIENT DIFFERENTIABLE NEURAL ARCHITECTURE SEARCH WITH MODEL PARALLELISM

## ABSTRACT

Neural architecture search (NAS) automatically designs effective network architectures. Differentiable NAS with supernets that encompass all potential architectures in a large graph cuts down search overhead to few GPU days or less. However, these algorithms consume massive GPU memory, which will restrain NAS from large batch sizes and large search spaces (e.g., more candidate operations, diverse cell structures, and large depth of supernets). In this paper, we present binary neural architecture search (NASB) with consecutive model parallel (CMP) to tackle the problem of insufficient GPU memory. CMP aggregates memory from multiple GPUs for supernets. It divides forward/backward phases into several sub-tasks and executes the same type of sub-tasks together to reduce waiting cycles. This approach improves the hardware utilization of model parallel, but it utilizes large GPU memory. NASB is proposed to reduce memory footprint, which excludes inactive operations from computation graphs and computes those operations on the fly for inactive architectural gradients in backward phases. Experiments show that NASB-CMP runs $1.2\times$ faster than other model parallel approaches and outperforms state-of-the-art differentiable NAS. NASB can also save twice GPU memory more than PC-DARTS. Finally, we apply NASB-CMP to complicated supernet architectures. Although deep supernets with diverse cell structures do not improve NAS performance, NASB-CMP shows its potential to explore supernet architecture design in large search space [1].

## 1 INTRODUCTION

Neural architecture search (NAS) has revolutionized architecture designs of deep learning from manually to automatically in various applications, such as image classification (Zoph & Le, 2016) and semantic segmentation (Liu et al., 2019a). Reinforcement learning (Zoph & Le, 2016; Zoph et al., 2018; Pham et al., 2018), evolutionary algorithms (Real et al., 2017; 2019), and differentiable algorithms (Liu et al., 2019b; Cai et al., 2019) have been applied to discover the optimal architecture from a large search space of candidate network structures. Supernets (Zoph et al., 2018; Pham et al., 2018) comprising all possible networks reduce search spaces from complete network architectures to cell structures. Recent acceleration techniques of differentiable NAS (Xie et al., 2019; Yao et al., 2020; Chen et al., 2019; Xu et al., 2020) further diminish search costs to affordable computation overheads (e.g., half GPU day). Prior work (Xu et al., 2020) randomly samples partial channels of intermediate feature maps in the mixed operations.

However, supernets of differentiable NAS consume gigantic GPU memory, which constrains NAS from using large batch sizes and imposes restrictions on supernet architectures' complexity. For example, NAS determines networks in shallow supernets (e.g., 8 layers) for deep compact networks (e.g., 20 layers). The cell structures are also required to remain identical for the same type of cells. Data parallelism can increase the search efficiency of NAS by using large batch sizes, such as SNAS (Xie et al., 2019), but it requires supernet complexity low enough to fit in a single GPU. In contrast, model parallelism can parallelize complex supernets, which distributes partial models to multiple devices. Nevertheless, model parallelism suffers from low hardware utilization. Only one device executes its model partition, while other devices stay idle. How to take advantage of multiple GPUs for large supernets efficiently is an open problem.

---

[1]Search and evaluation code are released at link

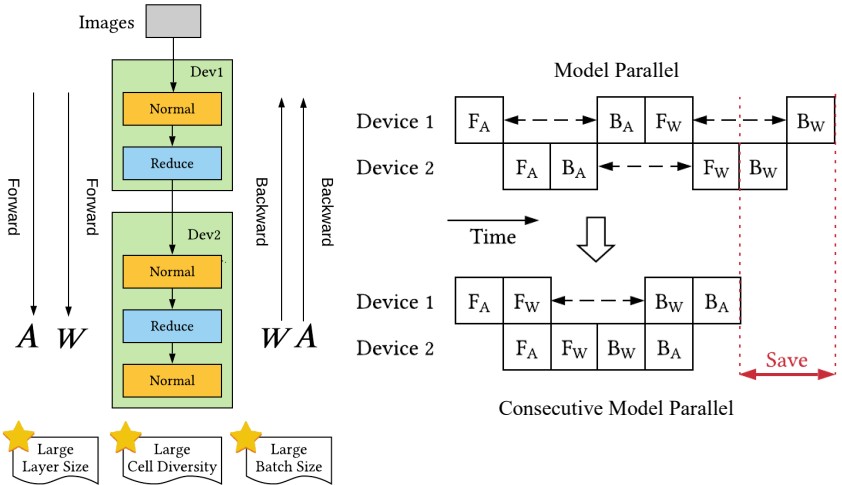

Figure 1: Consecutive model parallel (CMP) overlaps the two forward sub-tasks ($F_A$ and $F_W$) and two backward sub-tasks ($B_W$ and $B_A$). This new execution order empowers neural architecture search (NAS) to search faster than using model parallel (MP). The right figure shows that CMP can save two cycles from vanilla MP. Furthermore, CMP inherits MP's advantages, like using large batch sizes in the supernet, enlarging layer numbers of the supernet, and even diversifying cell architecture across different layers.

In this paper, we propose a simple and efficient solution, binary neural architecture search (NASB) using consecutive model parallel (CMP), to tackle the above limitations. Specifically, supernets have two forward and two backward phases to learn architecture parameters and network weights. CMP distributes several sub-tasks split from the four phases in multiple GPUs and executes the sub-tasks of all forward/backward phases together. Figure 1 illustrates that sub-tasks of forward/backward phases will be overlapped to reduce waiting cycles. Nevertheless, CMP consumes large GPU memory due to two computation graphs existing at the same time. Thus, we introduce NASB to declines GPU memory occupation. NASB utilizes binary and sparse architecture parameters (1 or 0) for mixed operations. It excludes inactive operations in the computation graph and computes feature maps of inactive operations for architecture gradients during the back-propagation. In this way, NASB-CMP can increase hardware utilization of model parallelism with efficient GPU memory in differentiable NAS.

In our experiments on CIFAR-10, NASB-CMP runs $1.2\times$ faster than using model parallel and pipeline parallel, TorchGPipe (Kim et al., 2020) in a server with 4 GPUs [2]. It can achieve the test error of $2.53 \pm 0.06\%$ by searching for only 1.48 hours. Our contribution can be summarized as follows:

- NASB-CMP is the first NAS algorithm that can parallelize large supernets with large batch sizes. We analyze the acceleration ratio between CMP and traditional model parallelism. Even though complex supernets (e.g., large layers and different cell structures) will not boost NAS performance, NASB-CMP paves the way to explore the supernet architecture design in the future.

- NASB utilizes binary architecture parameters and extra architecture gradients computation to reduce GPU usage. It can save memory consumption by accepting twice batch sizes larger than the other memory saving algorithm, PC-DARTS (Xu et al., 2020).

- We fairly compare NASB-CMP with state-of-the-art differentiable NAS in the same hardware and search space. Extensive experiments show that NASB-CMP can achieve competitive test error in short search time.

---

[2]NVIDIA GTX 1080 Ti.

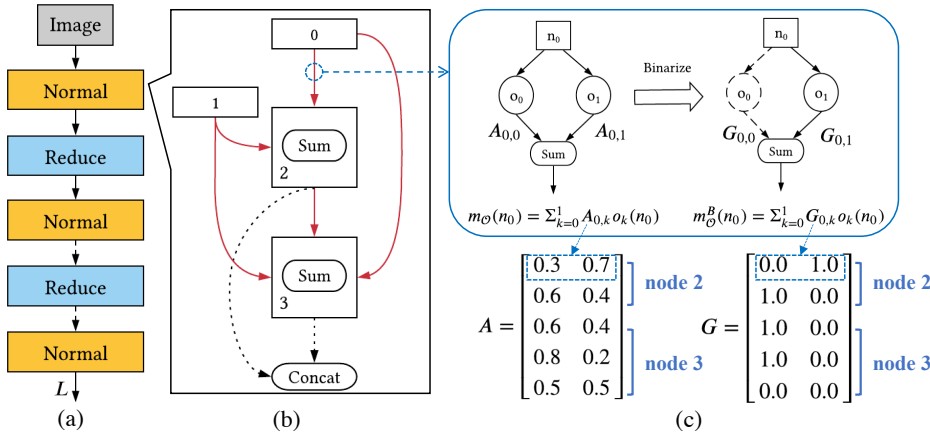

Figure 2: Illustration of Binary Neural Architecture Search (NASB). (a) is a supernet made up of normal and reduce cells. (b) portrays the directed-acyclic-graph (DAG) used for cell structures. (c) embodies the mixed operation in the solid red lines of the middle figure. NASB builds its supernet with binary mixed operations $m_{\mathcal{O}}^B$, which replace architectural matrix $\boldsymbol{A}$ with binary matrix $\boldsymbol{G}$. The symbol $n$ and $o$ stand for nodes in DAG and candidate operations. Among rows associated with a node (blue bracket) in $\boldsymbol{A}$, the largest two values are set to 1 and the rest elements to 0. Only partial operations are active during the search procedure of NASB.

## 2 METHODOLOGY

We first describe the fundamental concepts of one-shot neural architecture search (NAS) in Section 2.1. We then portray the consecutive model parallel to enhance NAS search efficiency in multiple devices in Section 2.2. Finally, we explain how we binarize the architectural weights and compute their gradients to cut down the GPU memory consumption in Section 2.3.

### 2.1 ONE-SHOT NEURAL ARCHITECTURE SEARCH

One-shot neural NAS (Zoph et al., 2018) is built on a supernet (a.k.a. meta graph) in which we stack normal cells and reduce cells sequentially in Figure 2 (a). Normal cells are analogous to convolutional layers to extract images features. Reduce cells are equivalent to pooling layers to reduce the spatial dimension of feature maps. All normal cells share the same structure, but each cell still has its network weights. So do all reduce cells. One-shot approaches are required to design two cell structures instead of complete neural networks. Figure 2 (b) illustrates one popular cell structure (Pham et al., 2018), an $N$-node directed-acyclic-graph (DAG) with total edges $E$, not counting the "concat" node. In the $h$-th cell, the first two nodes are the $(h-2)$-th and $(h-1)$-th cells having no inbound edges. The other nodes accept previous nodes whose index is lower than the current index. Total edges $E$ (red lines of Figure 2 (b)) is $(N+1)(N-2)/2$. We denote the $h$-th cell's output as $y_h = concat(n_j)$, where $2 \le j \le N-1$ and $n_j$ is a DAG node signified in Eq. 1.

$$n_j = \begin{cases} y_{h-2}, & \text{if } j = 0, \\ y_{h-1}, & j = 1, \\ \sum_{i<j} m_{\mathcal{O}}(n_i), & 2 \le j \le N-1. \end{cases} \tag{1}$$

A mixed operation $m_{\mathcal{O}}$ is the edge between node $i$ and $j$ in the DAG. Let $\mathcal{O}$ be a set of candidate operations (e.g., convolution, pooling, identity, zero) and $\boldsymbol{A} \in \mathbb{R}^{E \times |\mathcal{O}|}$ be a matrix of architecture parameters. Eq. 2 formulates the mixed operation $m_{\mathcal{O}}$ from node $i$ to $j$ as the weighted sum of all operations $o_k$ (Liu et al., 2019b).

$$m_{\mathcal{O}}^j(n_i) = \sum_{k=1}^{|\mathcal{O}|} \boldsymbol{A}_{e,k} o_k(n_i), \ j \ge 2, \ i < j, \tag{2}$$

where $e = (j+1)(j-2)/2 + i$ is the edge index. The mixed operations transform the cell structure search to the problem of learning two matrices, $\boldsymbol{A}_N$ and $\boldsymbol{A}_R$, for the normal and reduce cell.

Given that $\mathcal{L}_{val}$ and $\mathcal{L}_{train}$ is the loss function $\mathcal{L}$ beyond a training and validation dataset, respectively. Let $\mathbf{A}$ comprise $\mathbf{A}_N$ and $\mathbf{A}_R$. Mathematically, one-shot NAS can be formulated in the following optimization problem,

$$min_{\mathbf{A}} \ \mathcal{L}_{val}(w^*, \mathbf{A})$$
$$s.t. \ w^* = \arg\min_{w} \mathcal{L}_{train}(w, \mathbf{A}). \tag{3}$$

NAS leverages the validation performance to choose well-trained networks that outperform others. After training $\mathbf{A}$, we derive the compact network by pruning unused operations in the supernet. Since the whole paper follows the image classification setting (Liu et al., 2019b; Cai et al., 2019), we assume each node is assigned two inputs and two operations. And we prune node inputs of cells of the supernet by the largest two values of $\mathbf{A}$ associated with that node. For simplicity, we use $A$ in replace of $\mathbf{A}$ in the following discussion.

## 2.2 CONSECUTIVE MODEL PARALLEL

Data parallelism can scale up supernets with large batch sizes, but it cannot handle large supernets (e.g., deep supernets with different cell structures). Model parallelism (MP) is able to amortize such large supernets across multiple GPUs, but its hardware utilization is low. MP would generate unwanted waiting cycles across devices. Figure 1 displays that the first device becomes idle until the second device finishes its forward and backward phases. The parallelization gets worse as we use large available GPUs.

Motivated by pipeline parallelism (Huang et al., 2019), we propose consecutive model parallel (CMP) to decrease GPU idle time. Let $F_{\mathbf{A}}$ and $B_{\mathbf{A}}$ signify the forward and backward phase to update $\mathbf{A}$, and $F_w$ and $B_w$ be two phases to update $w$. CMP divides the four phases into several sub-tasks and performs sub-tasks of $F_{\mathbf{A}}$ and $F_w$ consecutively, followed by sub-tasks of $B_w$ and $B_{\mathbf{A}}$. Figure 1 illustrates that the execution order change by CMP overlaps sub-tasks without waiting for others to finish. Given the number of available GPUs $M$, Eq. 4 reveals the ratio of execution time between CMP and MP in theory.

$$\frac{\text{Time of CMP}}{\text{Time of MP}} = \frac{\frac{1}{M}[4M - 2(M-1)]}{4} = 1 - \frac{M-1}{2M}. \tag{4}$$

We assume $F_{\mathbf{A}}$, $B_{\mathbf{A}}$, $F_w$, and $B_w$ take the same time unit. MP will complete an iteration in 4 units. For CMP, the total sub-tasks is $4M$, and $2(M-1)$ sub-tasks can be overlapped. If a sub-task takes $1/M$ ideally, CMP will finish an iteration in $1/M(4M - 2(M-1))$ units. According to Eq. 4, CMP with two devices could reduce (2-1)/(2*2)=25% time from MP. In practice, Experiment 3.1 demonstrates that NASB-CMP runs $1.2\times$ faster than model parallelism without sacrificing test error. The theoretical value for 4 GPU is 1.6 (or reduce 37.5% time). We believe communication overhead and uneven model balance cause the deviation. Communication overhead comes from the intermediate tensors transfer from one to another GPU when models are split into different GPUs. Moreover, the main thread is responsible for loading data and backward propagation. The GPU with the main thread always consumes the most GPU memory, which causes uneven model balance.

CMP is a general model parallel approach for any existing differentiable NAS algorithm. However, running $B_{\mathbf{A}}$ and $B_w$ consecutively asks for two computation graphs, which doubles GPU utilization and deteriorates CMP efficiency. To address the problem of great GPU consumption, we introduce a memory-efficient NAS to CMP, called binary neural architecture search (NASB).

## 2.3 BINARY NEURAL ARCHITECTURE SEARCH

Binary neural architecture search (NASB) harnesses binary mixed operations $m_{\mathcal{O}}^B$ (Yao et al., 2020) that convert the real-valued $A$ into sparse binary matrix $G$, as illustrated in Figure 2. Among rows $A_{e,:}$ associate node $j$, $m_{\mathcal{O}}^B$ enforces the two largest elements to 1 (active) and the rest elements to 0 (inactive). The row indexes of active elements indicate selected edges to node $j$, while column indexes indicate chosen operations. Notice that NASB does not directly multiply $G$ with candidate operations in Eq. 5. Instead, NASB constructs a set of active operations $\mathcal{O}^{(active)}$ based on active elements in $G$. Only those active operations $o_a \in \mathcal{O}^{(active)}$ are included in the forward phase. This technique could stop inactive operations being stored in the computation graph and decrease roughly

---

**Algorithm 1:** NASB - Consecutive Model Parallel

---
1: Initialize architecture weights $\boldsymbol{A}$ and network weights $w$
2: **while** not stopped **do**
3:    $\boldsymbol{G}_t = binarize(\boldsymbol{A}_t)$
4:    Create $m_{\mathcal{O}}^B$ using $\boldsymbol{G}_t$ and Eq. 5
5:    Compute $\mathcal{L}_{valid}(w_t, \boldsymbol{G}_t)$ and $\mathcal{L}_{train}(w_t, \boldsymbol{G}_t)$ consecutively // model parallel
6:    Compute $\nabla_w \mathcal{L}_{train}(w_t, \boldsymbol{G}_t)$ and $\nabla_{\boldsymbol{A}} \mathcal{L}_{valid}(w_t, \boldsymbol{G}_t)$ consecutively // model parallel
7:    Update $w_{t+1}$ by descending $\nabla_w \mathcal{L}_{train}(w_t, \boldsymbol{G}_t)$
8:    Update $\boldsymbol{A}_{t+1}$ by descending $\nabla_{\boldsymbol{A}} \mathcal{L}_{valid}(w_t, \boldsymbol{G}_t)$
9: **end while**

---

$|\mathcal{O}|$ times GPU memory compared to using the multiplication by $\boldsymbol{G}$.

$$m_{\mathcal{O}}^B(n_i) = \sum_{k=1}^{|\mathcal{O}|} \boldsymbol{G}_{e,k} o_k(n_i) = o_a(n_i). \tag{5}$$

NASB computes gradients of network weights $w$ using standard back-propagation in the supernet. For the gradients of $\boldsymbol{A}$, NASB estimates $\partial \mathcal{L} / \partial \boldsymbol{A}$ approximately by $\partial \mathcal{L} / \partial \boldsymbol{G}$:

$$\frac{\partial \mathcal{L}}{\partial \boldsymbol{A}_{e,k}} = \frac{\partial \mathcal{L}}{\partial m_{\mathcal{O}}} \frac{\partial m_{\mathcal{O}}}{\partial \boldsymbol{A}_{e,k}} \approx \frac{\partial \mathcal{L}}{\partial m_{\mathcal{O}}^B} \frac{\partial m_{\mathcal{O}}^B}{\partial \boldsymbol{G}_{e,k}} = \frac{\partial \mathcal{L}}{\partial m_{\mathcal{O}}^B} \times o_k(n) = \frac{\partial \mathcal{L}}{\partial \boldsymbol{G}_{e,k}}. \tag{6}$$

Eq. 6 states that gradients of elements in $\boldsymbol{A}$ come from $\partial \mathcal{L} / \partial m_{\mathcal{O}}^B \times o_k(n)$. However, inactive operations are not in the computation graph. NASB saves inputs of inactive operations $n$ in PyTorch Context that is used for backward computation. During the backward phase, NASB will compute inactive operations $o_{k'}(n)$ on the fly and multiply the results with the $\partial \mathcal{L} / \partial m_{\mathcal{O}}^B$.

Apart from saving unneeded GPU FLOPS and memory, $m_{\mathcal{O}}^B$ can avoid performance bias between supernets and compact networks. Supernets using $m_{\mathcal{O}}$ assume that the performance of supernets can represent derived compact networks, but non-linear operations (e.g., ReLU-Conv-BN) break the representation that causes performance bias (Xie et al., 2019). Instead, the sparse matrix of $m_{\mathcal{O}}^B$ activates one operation. The performance of supernets during the search is only for one compact network. Thus, NASB can mitigate the bias caused by non-linear operations.

Algorithm 1 describes how CMP works with NASB. Note that NASB-CMP does not update any parameter (including $\boldsymbol{A}$ and $w$) until $F_{\boldsymbol{A}}$, $B_{\boldsymbol{A}}$, $F_w$, and $B_w$ complete. $\mathcal{L}_{train}$ will use the current binary architecture matrix $G_t$ rather than updated $G_{t+1}$, which is the major difference from the alternate algorithm (See Appendix A). Experiment 3.2 demonstrates NASB could save substantial GPU memory than PC-DARTS (Xu et al., 2020), which reduces GPU memory by partial channels of feature maps in mixed operations.

**Comparison with other methods**. NASP (Yao et al., 2020) binarizes $\boldsymbol{A}$ based on $\boldsymbol{A}$ itself, while ProxylessNAS (Cai et al., 2019) binarizes $\boldsymbol{A}$ based on the softmax results of $\boldsymbol{A}$. The two binarization approaches are equivalent, but how they handle binary mixed operations (Eq. 5) is different. NASP multiplies $\boldsymbol{G}$ with all operations (i.e., saving active and inactive operations in the computation graph). ProxylessNAS selects two sampled operations (paths) in the computation graph according to multinomial distribution. NASB utilizes the same binarization as NASP but only keeps one active operation in the computation graph according to $\boldsymbol{G}$.

## 3 EXPERIMENTS

We compare NASB-CMP with other parallelisms on the CIFAR-10 in Section 3.1. We then inspect the quality of NASB and compare NASB-CMP with state-of-the-art NAS in Section 3.2. Finally, we investigate the design of supernet architectures using large layers and different cell structures in Section 3.3, which cannot be conducted without saving GPU consumption or model parallel.

**Dataset.** CIFAR-10 Krizhevsky & Hinton (2009) is a color-image dataset for image classification, composed of 50,000 training images and 10,000 test images for 10 classes. The dataset preparation can be found in Append E.

**Search Space.** The DAG (See Section 2.1) has $N = 6$ intermediate nodes and $E = 14$ total edges. The set of candidate operations follows NASP (Yao et al., 2020), where normal operations $|\mathcal{O}_N|=8$ and reduce operations $|\mathcal{O}_R|=5$. Notice that our baselines also use the same operation sets rather than their original one ($|\mathcal{O}_N| = |\mathcal{O}_R| = 8$). All operations are included in Appendix F.

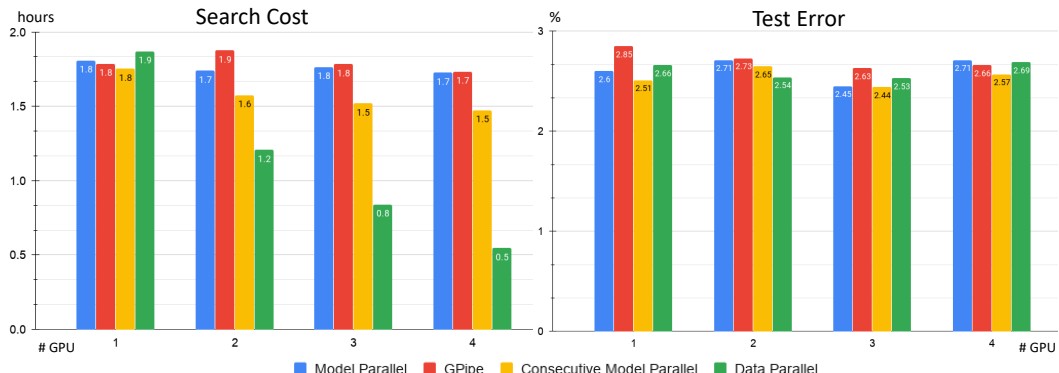

Figure 3: Performance comparison between different parallel approaches in NAS (best viewed in color). GPipe (Huang et al., 2019) is an approach of pipeline model parallel. Among the three model parallel approaches (blue, red, yellow), consecutive model parallel (CMP) outperforms them in terms of search cost and test error (lower is better). While the data parallel (green) is the fastest parallel method, but its test errors are not as low as CMP.

### 3.1 PARALLELISM COMPARISON ON CIFAR-10

The performance of NASB-CMP is compared with other parallel approaches on CIFAR-10, including data parallelism, model parallelism, and GPipe (Huang et al., 2019), the state-of-the-art model parallel that pipelines chunks of data into several model partitions. The implementation of Parallel NASB is written in Appendix B, and the search and evaluation settings are in Appendix C and D.

Figure 3 compares the performance of different parallelizations in NASB in varied GPUs. CMP runs $1.2\times$ faster than model parallel (MP) and GPipe especially running in 3 and 4 GPUs. According to Eq. 4, four GPUs should run 1.6X faster (or reduce 37.5% search time) than MP. In practice, communication overhead and uneven model partitions reduce the ideal speedup ratio. Compared with all parallel approaches, CMP's execution order change does not degrade the test error. Data parallel takes the lowest search cost, but it does not generate as low test error as other model parallel approaches. The reason might be that model replicas in data parallel utilize partial batches to compute architectural gradients, while model parallel can make use of the whole batches. Therefore, CMP is an efficient model parallel approach that helps NAS to utilize large batches.

Despite the competitive performance, the scalability of CMP is inferior. CMP disallows batch sizes from linearly scaling up as large GPUs are involved. For example, 2 GPUs should use 448 (we used 416 instead) if 1 GPU uses 224. Besides, 1-GPU NASB can utilize batch size 448, but NASB-CMP needs 4 GPUs to double batch sizes. The main reason is that CMP keeps two computation graphs (for $B_A$ and $B_w$) simultaneously for overlapping computations, resulting in twice GPU consumption. We believe that a mixed parallel combining CMP and data parallel can mitigate the drawback by merging two advantages, accuracy of CMP and scalability of data parallel.

### 3.2 STATE-OF-THE-ART NAS COMPARISON ON CIFAR-10

Following experiment settings in Appendix C and D, we compare NASB and NASB-CMP with several NAS algorithms on CIFAR-10. DARTS (Liu et al., 2019b), SNAS (Xie et al., 2019), NASP (Yao et al., 2020), and PC-DARTS (Xu et al., 2020) are selected as our baselines. DARTS is the pioneer of differentiable NAS. SNAS points out the performance bias between a supernet and derived networks in DARTS. Both NASP and PC-DARTS reduce GPU memory, which overlaps the scope of this paper. We should select ProxylessNAS (Cai et al., 2019) as a baseline, but their search code on CIFAR-10 is not released. We prefer not to ruin their performance with improper implementation.

Table 1: Comparison with state-of-the-art NAS on CIFAR-10

| Model | Test Error (%) | Params (M) | Search Cost (GPU hours) | Search Batch Size |
|---|---|---|---|---|
| DenseNet-BC Huang et al. (2017) | 3.46 | 25.6 | - | - |
| NASNet-A + c/o (Zoph et al., 2018) | 2.65 | 3.3 | 43200 | - |
| AmoebaNet-B + c/o (Real et al., 2019) | $2.55 \pm 0.05$ | 2.8 | 75600 | - |
| ENAS + c/o (Pham et al., 2018) | 2.89 | 4.6 | 12 | - |
| ProxylessNAS-G + c/o (Cai et al., 2019) | 2.08 | 5.7 | - | - |
| NAONet-WS + c/o (Luo et al., 2018) | 2.93 | 2.5 | 7.2 | - |
| AlphaX + c/o (Wang et al., 2019) | 2.06 | 9.36 | 360 | - |
| DARTS (2nd order) + c/o (Liu et al., 2019b) | $2.83 \pm 0.06$ | 3.4 | 96 | 64 |
| SNAS-moderate + c/o (Xie et al., 2019) | $2.85 \pm 0.02$ | 2.8 | 36 | 64 |
| NASP + c/o (Yao et al., 2020) | $2.44 \pm 0.04$ | 7.4 | 4.8 | 64 |
| PC-DARTS + c/o (Xu et al., 2020) | $2.57 \pm 0.07$ | 3.6 | 2.4 | 256 |
| DARTS (2nd order) + c/o (Liu et al., 2019b) | $7.25 \pm 4.20$ | $1.8 \pm 0.6$ | $53.62 \pm 5.06$ | 60 |
| SNAS + c/o (Xie et al., 2019) | $\mathbf{2.58 \pm 0.08}$ | $8.8 \pm 0.5$ | $11.06 \pm 0.2$ | 60 |
| NASP + c/o (Yao et al., 2020) | $2.76 \pm 0.35$ | $5.5 \pm 0.8$ | $6.44 \pm 0.12$ | 60 |
| PC-DARTS + c/o (Xu et al., 2020) | $2.59 \pm 0.05$ | $6.5 \pm 0.9$ | $8.96 \pm 0.08$ | 60 |
| NASB + c/o | $2.64 \pm 0.09$ | $5.4 \pm 1.2$ | $\mathbf{3.92 \pm 0.38}$ | 60 |
| PC-DARTS + c/o (Xu et al., 2020) | $2.60 \pm 0.17$ | $5.5 \pm 1.2$ | $4.10 \pm 0.03$ | 224 |
| NASB + c/o | $\mathbf{2.49 \pm 0.07}$ | $6.9 \pm 1.5$ | $1.64 \pm 0.06$ | 448 |
| NASB + CMP + c/o | $2.53 \pm 0.06$ | $6.9 \pm 0.7$ | $(\mathbf{1.48 \pm 0.02}) \times 4$ | 896 |

Instead of directly using their reported results, we re-run the baselines from scratch to ensure their hardware and search space are the same. So, we can fairly compare them in terms of test error and search cost.

The test error and search cost on CIFAR-10 are stated in Table 1, where "c/o" signifies Cutout (De-Vries & Taylor, 2017) used in the evaluation phase. The first row (human designed networks) and the second group of rows are extracted from their papers. The third group compares NASB with differentiable NAS baselines. The fourth group compares NAS algorithms using large batch sizes. Notably, ProxylessNAS attains the outstanding test error, but its supernet structure and search space are different from what we use, which might bias the comparison.

In the third group, NASB significantly takes the cheapest search cost, roughly 4 hours, to reach a comparable test error of 2.64% with SNAS 2.58% and PC-DARTS 2.59%, not to mention the search cost is smaller than the second group. NASB and NASP use similar mixed binary operations, but NASB outperforms NASP in both search cost (3.92 versus 6.44) and test error (2.64 versus 2.76). The GPU memory utilization of NASB and NASP is 2,117 MB and 9,587 MB, respectively. These three comparisons indicate that the additional gradient computation for inactive operations is a useful technique. Notice that DARTS, SNAS, and PC-DARTS use different search space (See Appendix F), so the original test errors (second group) differ from what we report in the third group of Table 1. Especially, DARTS tends to overfit the validation set by selecting "skip_connect" in our search space. Its results are not as good as their original search space. Even though NASP uses the same search space, the different batch sizes and random seeds for the search and retrain setting still lead to different results.

The fourth group points out that NASB can considerably reduce GPU memory by using twice batch sizes larger than PC-DARTS within 1.64 hours to attain a test error of 2.49. PC-DARTS, how-ever, becomes worse when using large batch sizes. We consider that the Hessian approximation in PC-DARTS fluctuates greatly with large batch sizes, which misleads PC-DARTS to easily select "skip_connect". NASB-CMP using four GPUs enables twice batch sizes for NASB to finish its search in 1.48 hours without severe test error degradation. Its test error 2.53% also performs better than other differentiable NAS. The empirical results in the third and fourth groups demonstrate the high efficient NASB with significant memory saving and the strong performance of NASB-CMP.

## 3.3 LARGE SUPERNETS ON CIFAR-10

One-shot NAS embraces two limitations, (1) searching 8-layer supernets for 20-layer compact net-works and (2) same cell structures (Liu et al., 2019b; Pham et al., 2018). We hypothesize that

Table 2: Compare test error with different supernets on CIFAR-10

| Cell Structure | Same | | Different | |
|---|---|---|---|---|
| # Layers | 8 | 20 | 8 | 20 |
| NASB | 2.59 | 2.78 | 2.52 | 2.82 |
| NASB-CMP | 2.58 | 2.78 | 2.76 | 2.68 |

20-layer supernets with different cell structures can build suitable compact networks. Thanks to NASB and NASB-CMP that reduce GPU utilization and exploit multiple GPUs, we can examine how supernet architectures affect NAS. The search and retrain setting follow Appendix C and D. Table 2 shows test errors on CIFAR-10 with various supernet architectures, where the 1st and 2nd rows indicate cell diversity and layers (cells) numbers. Since 8-layer supernets could have six varying normal cells, we magnify each normal cell three times to construct compact networks.

First, supernets with large layers do not benefit NAS to discover high-quality compact networks. Test errors in NASB (3rd row) and NASB-CMP (4th row) show that most 8-layer supernets can generate lower test errors than 20-layer supernets. The reason is 20-layer supernets have numerous architecture parameters and network weights, and they should ask for more search epochs than 8-layer supernets to train. Insufficient search epochs for deep supernets do not help NAS reach strong compact networks. Furthermore, supernets with different cell structures are not beneficial for NAS as well. When we compare results in 2nd and 4th columns (or 3rd and 5th columns), most supernets using the same cell structures can generate similar or lower test errors than using different cell structures. The reason is close to the previous one. Different cell structures demand extra search epochs to train high-dimensional architecture parameters compared to homogeneous cell structures. Not enough epochs for different cell structures do not produce low test error. Although the results contradict the hypothesis, NASB-CMP shows its potential to explore complicated supernet architectures, which paves the way for designing supernet architectures.

## 4 RELATED WORK

Parallelism has been applied to NAS for acceleration (Zoph & Le, 2016; Xie et al., 2019; Cai et al., 2019; Mei et al., 2019). Parameter servers in NAS (Zoph & Le, 2016) train several child networks in parallel to speed up the learning process of the controller. ProxylessNAS (Cai et al., 2019) speed up its retrain phase by a distributed framework, Horovod (Sergeev & Del Balso, 2018). SNAS (Xie et al., 2019) and AtomNAS (Mei et al., 2019) have accelerated the search phase by data parallelism. Data parallelism runs data partitions simultaneously across multiple devices, but it cannot parallelize large models exceeding the memory of a single device, especially complicated supernets with large batch sizes. In contrast, model parallelism (Lee et al., 2014; Harlap et al., 2018; Huang et al., 2019; Kim et al., 2020) excels at parallelizing large models. GPipe (Huang et al., 2019) splits mini-batches to micro-batches and execute micro-batches in the pipeline of model partitions. The pipeline manner mitigates low hardware utilization in model parallelism. Consecutive model parallel is motivated by pipeline parallelism to overlap sub-tasks of forward/backward phases. We found that batch splitting and re-materialization (Chen et al., 2016) of GPipe increase NAS search time because frequently updating $A$ and $w$ intensifies extra computation. To the best of our knowledge, CMP is the most efficient model parallelism for NAS.

Reducing GPU utilization to enlarge search batch sizes is another acceleration techniques (Xu et al., 2020; Chen et al., 2019; Xie et al., 2019; Yao et al., 2020; Cai et al., 2019). PC-DARTS (Xu et al., 2020) samples channels of feature maps in mixed operations. P-DARTS (Chen et al., 2019) reduce search space as it progressively increases layers of supernets in the search phase. ProxylessNAS (Cai et al., 2019) and NASP (Yao et al., 2020) binarize $A$ to reduce all operations saved in GPU. NASB uses the same binarization as NASP but saves one active operation in the mixed operations. Thus, NASB can reduce GPU consumption substantially and give CMP more space to keep two computation graphs in GPUs.

## 5 CONCLUSION

We proposed a simple and efficient model parallel approach, NASB-CMP, which overlaps sub-tasks of forward and backward phases to reduce idle time across GPUs and utilize binary architecture parameters to reduce GPU utilization for heavy supernets. Experiments on CIFAR-10 show NASB-CMP runs $1.2\times$ faster with a large batch size of 896 than other model parallel approaches in 4 GPUs and only took 1.48 hours to attain a test error of 2.53, surpassing state-of-the-art differentiable NAS. Moreover, NASB-CMP is able to accommodate high complicated supernets for search, which paves the way for supernet network architecture design. In the future, we will combine the data parallel with NASB-CMP to overcome its inferior scalability, investigate effective and complicated supernet architectures, and analyze the communication overhead of NASB-CMP in a multi-node GPU cluster.

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

## A    ALTERNATE ALGORITHM OF BINARY NEURAL ARCHITECTURE SEARCH

Algorithm 2 displays the alternate fashion to update $A$ and $w$ in the NAS: updating $A$ with fixed $w$ and then updating $w$ with fixed $A$. Note that Line 9 of Algorithm 2 computes the gradients $\nabla_w \mathcal{L}_{train}$ with updated $G_{t+1}$, which is different from the consecutive algorithm (Algorithm 1).

---

**Algorithm 2:** NASB

---

1:  Initialize architecture weights $A$ and network weights $w$
2:  **while** not stopped **do**
3:     $G_t = binarize(A_t)$
4:     Create $m_{\mathcal{O}}^B$ using $G_t$ and Eq. 5
5:     Compute $\nabla_A \mathcal{L}_{valid}(w_t, G_t)$ using Eq. 6 // handle the gradients of inactive elements
6:     Update $A_{t+1}$ by descending $\nabla_A \mathcal{L}_{valid}(w_t, G_t)$
7:     $G_{t+1} = binarize(A_{t+1})$
8:     Create $m_{\mathcal{O}}^B$ using $G_{t+1}$ and Eq 5
9:     Compute $\nabla_w \mathcal{L}_{train}(w_t, G_{t+1})$ // standard back-propagation
10:    Update $w_{t+1}$ by descending $\nabla_w \mathcal{L}_{train}(w_t, G_{t+1})$
11: **end while**

---

## B    IMPLEMENTATION OF PARALLEL NAS

The data parallel leverages PyTorch (Paszke et al., 2017) *distributed* module providing communication interfaces to update parameter tensors between multiple processes. Model parallel and CMP are implemented in multi-threading. Each GPU has a specialized thread responsible for its model partition. Those threads enable different model partitions to run simultaneously. Without multi-threading, only assigning model partitions to specific devices do not automatically overlap sub-tasks. For GPipe, we adopt the corresponding PyTorch package, torchgpipe (Kim et al., 2020), in replace of GPipe, since GPipe is written in Tensorflow. The chunk setting to split mini-batch size to micro-batch size is disabled in the experiment, because enabling the setting increases the search cost.

## C    SEARCH DETAILS ON CIFAR-10

Our platform is a server with 4 GPUs of NVIDIA GTX 1080 Ti, in which all search experiments are executed. Supernets consist of 8 cells in which the 3rd and 6th cells are reduce cells, and others are normal cells with initial channels 16. The optimizer for network weights $w$ is momentum SGD with moment 0.9, L2 penalty $3e - 4$, and cosine anneal learning rate initialized by 0.025 and minimal 0.001. The optimizer for architecture parameter $A$ is Adam with learning rate $3e - 4$, L2 penalty $1e - 3$, and $(\beta 1, \beta 2) = (0.5, 0.999)$. PC-DARTS with large batch sizes (Xu et al., 2020) has unique configurations: initial learning rate 0.1 and minimal 0.0 for SGD optimizer and learning rate $6e - 4$ for Adam optimizer.

All NAS algorithms will search networks for 50 epochs with varied batch sizes and random seeds. In Experiment 3.1, NABS-CMP is specified search batch size 224, 416, 512, 896 for 1, 2, 3, 4 GPUs, respectively It random seed is 2. In Experiment 3.2, The batch size is 60 determined by DARTS because DARTS consumes the largest GPU memory. We want all NAS algorithms to use the same batch size in order to compare each other fairly. Since PC-DARTS is proposed to reduce GPU memory consumption, we also compare the performance of PC-DARTS using a large batch size 224 with NASB and NASB-CMP. NASB is specified with its allowable maximal batch size 448 in a single GPU, and NASB-CMP uses a batch size of 896 in 4 GPUs. All NAS baselines and NASB use 2, 3, 4, 5, 6 as random seeds, and NASP-CMP uses 2, 3, 9, 11, 18 instead. In Experiment 3.3, NASB and NASB-CMP exploit batch size 160 and 256, respectively, for 50 epochs. We ran search experiments twice using random seed 2 and 3 and reported the average test error among the two searches in Table 2.

# D    EVALUATION DETAILS ON CIFAR-10

The compact networks used in the retrain (evaluation) phase have 20 cells (layers), where the one-third and two-thirds of the depth are reduce cells and others are normal cells. We retrain the compact networks from scratch for 600 epochs with the batch size 96, dropout path of probability 0.2, and initial channels of 36. We also add the auxiliary layer in the network with a loss weight 0.4. During the evaluation phase, the cutout length 16 is additionally applied for image transformation. The optimizer setting for network weights $w$ is the same as the searching setting. The retrain random seed is assigned to 0, which is different from the search seeds.

# E    PRE-PROCESSING SUMMARY ON CIFAR-10

We preprocess the training images in the following techniques: padding $32\times32$ images with 4 pixels, and then randomly cropping them back to $32 \times 32$; randomly flipping images in the horizontal direction; normalizing image pixels by the channel mean and standard deviation. The processed training set is split evenly: the first half serves as the final training set, and the other serves as the validation set. SNAS merely relies on the training set to search, so its training set is not split.

# F    CANDIDATE OPERATIONS ON CIFAR-10

Table 3 summarizes candidate operations for mixed operations used in NAS papers. Experiment 3 makes use of the first row of Table 3 as its search space on CIFRAR-10. "skip_connect" symbolizes identity operation if stride size is 1 or ReLU-Conv-Conv-BN operation. "conv", "sep", and "dil_conv" signifies convolution, depthwise-separable convolutions, and dilated depthwise-separable convolutions, respectively. "none" means the zero operation. Note that differentiable NAS baselines (DARTS, SNAS, PC-DARTS) also utilize the first row of Table 3 as their search space.

Table 3: Candidate operations for normal and reduce cells

|  | Normal Cell | Reduce Cell |
|---|---|---|
| NASB-CMP NASB NASP (Yao et al., 2020) | skip_connect (identity) conv_3x1_1x3 dil_conv_3x3 conv_1x1 conv_3x3 sep_3x3 sep_5x5 sep_7x7 | skip_connect (identity) avg_pool_3x3 max_pool_3x3 max_pool_5x5 max_pool_7x7 |
| DARTS (Liu et al., 2019b) SNAS (Xie et al., 2019) PC-DARTS (Xu et al., 2020) | none (zero) max_pool_3x3 avg_pool_3x3 skip_connect (identity) sep_3x3 sep_5x5 dil_conv_3x3 dil_conv_5x5 | |

