# OpenReview forum: "Efficient Differentiable Neural Architecture Search with Model Parallelism"
_ICLR.cc/2021/Conference — Reject_

### Official Review · AnonReviewer2 · 2020-10-26
**running batch for NAS supernet**

**Rating:** 6
**Confidence:** 4

**Review:**

##########################################################################
Summary:

The paper presents a method to run large batch size with supernets method for Neural Architecture Search (NAS).
It also shows model parallelization method CMP that allows 1.2x search speed improvement
Using a large batch improves the test accuracy because it allows the model to train on wider variety of data in the backpropagation.

##########################################################################

Reasons for score:

NASB-CMP algorithm that can use supernet with large batch and run faster
The paper shows model parallelization method CMP that allows 1.2x search speed improvement
The paper also shows that it improves the quality of the models by using this technique
The paper is clear and coherent

Testing the approach on other NAS Benchmarks (NAS-bench201) would be interesting.
Provide GPU system details that the experiments where ran on for the comparison with other methods.

##########################################################################
Pros:

- NASB-CMP algorithm that can use supernet with large batch and run faster

##########################################################################

Cons:

- Need to provide  GPU system details that the experiments where ran on for the comparison with other methods.
- Need to test on other NAS benchmarks

##########################################################################

Questions during rebuttal period:


Please address and clarify the cons above

---

> ### Author Response · Authors · 2020-11-18
> **We appreciate your comments.**
>
> **Need to provide GPU system details that the experiments where ran on for the comparison with other methods**
>
> Our experiment environment is a server made up of four GTX 1080 TI GPUs, twelve CPUs, and 128 GB main memory in which we search networks by NAS baselines, our proposed work, and data and model parallelism. Our software versions are python 3.6.3, pytorch 1.1.0, torchvision 0.3.0, and cuda 9.0. The rest of the search and evaluation setting could be found in Appendix C, D, and E.
>
> **Testing the approach on other NAS Benchmarks (NAS-bench201)**
>
> NAS-Bench 201 is constructed based on a 4-node DAG and five operations, which is smaller than our search space in the paper (6-node DAG, eight operations for normal cells, and five operations for reduce cells). So, we did not use the NAS-Bench201 benchmark.
>
> **Need to test on other NAS benchmarks**
>
> Apart from the CIFAR-10 datasets, the proposed NASB-CMP only took 7 hours to search networks on four GTX 1080 TI GPUs, while PC-DARTS needs to take 11.5 hours on eight Tesla V100 GPUs. This result demonstrates our search algorithm is also efficient in the large dataset. Currently, we are re-training the searched network from the ImageNet. If the results can finish before the rebuttal deadline, we will add them to the paper.

---

> ### Comment · Area_Chair1 · 2020-11-23
> **Feedback necessary**
>
> Dear reviewer,
>
> The authors have responded to your comments below. Could you please go over the response and give feedback to the authors sometime soon? The interactive discussion deadline is this Tuesday and you will not be able to interact with the authors after the date.
>
> Thanks,
> AC

---

### Official Review · AnonReviewer4 · 2020-10-26
**I believe the subject is important and I like the idea, but more experiments to demonstrate the effectiveness of the proposed approaches are needed.**

**Rating:** 5
**Confidence:** 3

**Review:**

##############################################################

Summary:

This paper provides the interesting method that leverages GPU memory resources more efficiently for supernet (meta-graph) of differentiable NAS. For this, this paper proposes binary neural architecture search and consecutive model parallel (CMP). CMP parallelizes one supernet with multiple GPUs, which allows NAS model to use larger batch size and search space. Additionally, this paper improves neural architecture search speed and hardware utilization with waiting cycles reduction by dividing forward/backward phases into several sub-tasks and executing the same type of sub-tasks. The proposed method shows 1.2x faster search time compared with other model parallel methods and the highest performance among differentiable NAS methods in the experiment section.

##############################################################

Reasons for score:

I vote for weak rejecting. I believe that the subject this paper handled is important and I like the idea which parallelizes the model (supernet) which has high potential of usefulness. My main concerns are the unclear experimental results for baseline models which hinders fair comparison and the lack of experiments for demonstrating the effectiveness of the proposed methods. During rebuttal period, I hope the authors can address my concerns.

##############################################################

Strong Points:

1. This paper tackles the main problems of the neural architecture search field: 1) search speed 2) resource efficiency 3) scalability of search space. I think the subject handled in this paper is timely and important.
2. This paper proposes a more improved binary neural architecture search than that which was proposed by the previous NAS method (NASP) in terms of both the neural architecture search performance and speed.
3. The proposed CMP has a lot of potentials to be useful and practical. The CMP is suitable to be applied to large-scale datasets and large-scale models. Also, since the CMP target differentiable NAS which is one of the most popular approaches in the NAS field, it has a probability to be developed as universally applicable tools for several differentiable NAS methods.

##############################################################

Weak Points:

1. Although this paper shows many experiments, I have some concerns about the reliability of the experimental results for the baseline models as follows.

(1) [PC-DARTS] I understand the search space in this paper is different from that of PC-DARTS, but all results components (test error, params, GPU hours) on CIFAR-10 of PC-DARTS in this paper are worse than those of PC-DARTS in the original paper as below. It is difficult to understand for me that the performance of PC-DARTS becomes worse even using more parameters. Could the authors report the search results on CIFAR-10 of the proposed method under the same search space of PC-DARTS?

Model                                      Test error(%)        Params(M)     GPU hours

PC-DARTS in the original paper: 2.57                      3.6                      2.4

PC-DARTS in this paper:               2.60                      5.5                     4.10

(2) [NASP] As I understand, this paper follows the search space of NASP. Then I think it is better to use the published results of NASP for a fair comparison.

Model                                                 Test error(%)        Params(M)    GPU hours

NASP (12 operation) in original paper: 2.44                         7.4                   4.8

NASP in this paper:                                   2.76                         5.5                  6.44

2. Since the batch sizes between NASP and NASB are different in Table 1, the effectiveness of each proposed binary neural architecture search is unclear for me. To clearly show the performance (e.g. the use of GPU memory, the speed gains, FLOPS) of the proposed binary neural architecture search (without CMP), could the authors show comparison between binary neural network search methods (The proposed model, NASP, ProxylessNAS) under the same batch size and a single GPU? (For ProxylessNAS, by referring CIFAR10 conditions in their paper with their official code)
3. I guess the proposed model will be effective for large-scale dataset such as ImageNet-1K. Could the authors validate the proposed method on ImageNet-1K?
4. Could the author give more information with specific values for the communication overhead and uneven model problems for better practical use of the proposed method?
5. Some expressions such as ‘harnesses’ give me the impression that this paper just use the binary neural architecture search of NASP. It would be better to focus to clarify the representation denoting the different points between them in the overall paper.
6. I believe if CMP is generalized to other differentiable NAS, CMP is very useful as a search time reduction tool for differentiable NAS even it does not achieve SOTA. Could the authors explain the direction of development of CMP to be integrated to other differentiable NAS such as DARTS, PC-DARTS?
7. Instead of model parallelism, if we parallelize the learning of architecture parameters and network weights as below concept, it seems to be reducing the search time more than CMP. (50% reduction with 2 GPUs)

Dev 1. F_A, F_A,  B_A, B_A    (Save?)

Dev 2. F_W,F_W, B_W, B_W<--------->

==> 4 phase

CMP is as follows:

Dev 1. F_A, F_W,<-------->B_W, B_A

Dev 2.      , F_A, F_W, B_W,B_A

==> 6 phase

(normal: 8phase)

Could you address my question by comparing those two parallelism methods?

8. I think it would be better to average multiple results with different seeds instead of picking top 1 from two results with two seeds.

##############################################################

Questions during rebuttal period:

Please address and clarify the weak points above.

##############################################################

Some typos:

1.2X --> 1.2$\times$

---
=====POST-REBUTTAL COMMENTS========

I thank the authors for the response and the efforts in the updated draft. Some of my queries were clarified. However, unfortunately, I still think more needs to be done to demonstrate the effectiveness of the proposed model on large dataset and analyze the effectiveness of each module (binary NAS and CMP). I keep my original decision for these reasons.

---

> ### Author Response · Authors · 2020-11-18
> **Thank you for the detailed comments the time.**
>
> **1. (1) It is difficult to understand for me that the performance of PC-DARTS becomes worse even using more parameters.**
>
> The search cost will positively correlate with the batch size and search space. In a GTX 1080 TI, our search space (NASP search space) prohibits us from using batch size 256 to run PC-DARTS, which is the major contribution to increase search cost in our results. Since the limited hardware resources and time constraints, searching networks on ImageNet and large supernets with more nodes in cell DAGs  (large models and large search space) are our priority. We could report the results of NASB and PC-DARTS in the PC-DARTS search space after the rebuttal period.
>
> **1. (2) Use the published results of NASP.**
>
> For the original results of NASP, we want to compare our algorithm with NASP in the same random seeds and the same GPU cards.
> If we directly use their results, the search setting and environment will be different from our work. So, we decided not to use the original results of NASP. By the way, we emphasize that our work uses the search space proposed by NASP.
>
> **2. NASB and NASP Performance Comparison.**
>
> In Table 1, we reported the search cost and test error of NASB and NASP with batch size 60. We can see NASB use fewer search time to achieve better test performance than NASP. For the GPU memory consumption with the same batch size 60, NASP would use 9,587 MB during the search phase. NASB would use at most 2,117 MB during the search phase. The low GPU consumption of NASB shows the effectiveness of our binary approach. For ProxylessNAS, the authors of ProxylessNAS do not release the search code for CIFAR-10. The search space and the superent of ImageNet are different from CIFAR-10. We feel sorry that we cannot compare ProxylessNAS binary ability with us.
>
> **3. Could the authors validate the proposed method on ImageNet-1K?**
>
> Apart from the CIFAR-10 datasets, the proposed NASB-CMP only took 7 hours to search networks on ImageNet with four GTX 1080 TI GPUs, while PC-DARTS needs to take 11.5 hours on eight Tesla V100 GPUs. This result demonstrates our search algorithm is also efficient in the large dataset. Currently, we are re-training the searched network from the ImageNet. If the results can finish before the rebuttal deadline, we will add the results to the paper.
>
> **4. The communication overhead and uneven model problems.**
>
> Currently, empirical results show that CMP speedup deviates from the theoretical values. Supernets are split into different GPUs. Communication overhead comes from intermediate tensors transfer from one to another GPU. Our experiments are executed in a single server (four GPU cards), so we do not have network communications. If one runs CMP in multiple servers, he/she should consider the network transmission overhead for tensors. Unfortunately, we do not have exact values for communication overhead so far. Uneven model balance among GPUs result from the main thread will handle data loading and backward propagation. We can observe that the GPU with main thread always consumes the most GPU memory and becomes the bottleneck of model parallelism.
>
> **7. New parallelize the learning of architecture parameters and network weights.**
>
> Our focus is model parallel. What you mention does not belong to model parallel. The theoretical speedup seems better than CMP, but your approach does not solve the large model issues in neural architecture search.
>
> We will update the typo in the rebuttal version. Thank you for pointing out.

---

> > ### Comment · AnonReviewer4 · 2020-11-23
> > **Thanks the authors for the response.**
> >
> > Thanks the authors for the response. After reading the response carefully, I feel that my concerns for 1-(2), 2, 3, 4 remain, and questions for 5, 6, 8 are not answered.
> >
> > Sorting comments by importance.
> >
> > ---
> >
> > **[Regarding 3]**
> >
> > I agree with the author's comments that the most important experiment to show the effectiveness of the proposed method is an experiment on a large-scale dataset such as ImageNet-1K. I really hope the authors upload the results before the 24th.
> >
> > ---
> >
> > **[Regarding 2]**
> >
> > Even NASP and NASB are compared in Table 1, they still not compared with the proposed binary neural architecture search. I want to see the effectiveness of the proposed binary neural architecture against NASP and NASB under the same batch size.
> >
> > ---
> >
> > **[Regarding 1]**
> >
> > (1) As you said, the different results of the original PC-DARTS and the reproduced PC-DARTS are not only originated from the different search space but the different environment settings. I recommend describing the environmental difference between them in the appendix.
> >
> > (2) I understand why the reproduced version of NASP is slower than the original. However, my concern for the test error remained since it is not affected by the type of graphic card. The original result 2.44 is outperformed the result of the proposed model in the main Table 1. I believe that considering the value of this paper, achieving SOTA is not important, but the main Table 1 should be revised.
> >
> > ---
> >
> > **[Regarding 4]**
> >
> > Could the authors give more analysis on why the empirical results have deviated from the theoretical values?
> >
> > ---
> >
> > Please answer the above mentioned additional concerns and 5, 6, 8. I understand that there is not enough time to respond. Please answer in order of importance in time possible.

---

> > > ### Author Response · Authors · 2020-11-24
> > > **Thank you for the further response.**
> > >
> > > **[Regarding 1] PC-DARTS and NASB original results**
> > >
> > > (1) We highlight the different batch sizes in Table 1 and different search space in Appendix F. Th experiment settings have been mentioned in the paper for PC-DARTS in Appendix C and D. That is why we do not write a separate section for different settings between the original one and ours.
> > >
> > > (2) Thank you for the suggestion. We have updated Table 1 with the original results of NASP, PC-DARTS, SNAS, and DARTS. We hope this table could help readers understand what performance SOTA are and where our proposed approach is. Notice that DARTS, SNAS, and PC-DARTS use different search space (See Appendix F), so the original test errors differ from what we report in Table 1. Even though NASP uses the same search space, the different batch sizes and random seeds for the search and retrain setting still
> > > lead to different results.
> > >
> > > **[Regarding 2] NASP and NASB comparison under the same batch size**
> > >
> > > NASB and NASP use similar mixed binary operations. Using the same batch size 60, NASB outperforms NASP in both search cost (3.92 versus 6.44) and test error (2.64 versus 2.76). The GPU memory utilization of NASB and NASP is 2,117 MB and 9,587 MB, respectively. These three comparisons indicate that the additional gradient computation (NASB) for inactive operations is a useful technique. We hope that this response can resolve your concern.
> > >
> > > **[Regarding 3] ImageNet-1K**
> > >
> > > The evaluation on ImageNet is proceeding to 20%. We feel really sorry that the test error will not be produced on time.
> > > Currently, the proposed NASB-CMP only took 7 hours to search networks on four GTX 1080 TI GPUs, while PC-DARTS needs to take 11.5 hours on eight Tesla V100 GPUs.
> > >
> > > **[Regarding 4] Empirical results have deviated from the theoretical values**
> > >
> > > To our best knowledge, we believe communication overhead and uneven model balance cause the deviation. Communication overhead comes from the intermediate tensors transfer from one to another GPU when models are split into different GPUs. Moreover, the main thread is responsible for loading data and backward propagation. The GPU with the main thread always consumes the most GPU memory, which causes uneven model balance.
> > >
> > > We will answer 5, 6, 8 in another comments. Thank you very much for the extensive discussion.

---

> ### Comment · Area_Chair1 · 2020-11-23
> **Feedback necessary**
>
> Dear reviewer,
>
> The authors have responded to your comments below. Could you please go over the response and give feedback to the authors sometime soon? The interactive discussion deadline is this Tuesday and you will not be able to interact with the authors after the date.
>
> Thanks,
> AC

---

> ### Author Response · Authors · 2020-11-24
> **Response for 5, 6, 8**
>
> **5. It would be better to clarify the representation between NASB and NASP in the overall paper.**
>
> In fact, NASB inherits the binary operations of NASP with gradient improvement to save a lot GPU memory consumption. Equation (6) describes how NASB to compute architecture gradients. We also described the difference between NASP and NASB in Section 2.3, especially, how we handle binary mixed operations. We did not want to deny the relationship between NASP and NASB and believed the description in Section 2.3 is enough to distinguish the difference between NASP and NASB. The experiments also shown NASB can outperform NASP in terms of accuracy, search cost, and GPU memory utilization.
>
> **6. Could the authors explain the direction of development of CMP to be integrated to other differentiable NAS such as DARTS, PC-DARTS?**
>
> The consecutive model parallel could be directly applied to DARTS and PC-DARTS to enable them to search with large batch size. DARTS and PC-DARTS also share the same execution flow (F_A, B_A, F_W, B_W). It should be easy to integrate CMP with DARTS and PC-DARTS.
>
> **8. It would be better to average multiple results with different seeds instead of picking top 1 from two results with two seeds.**
> You provide a good suggestion. We will use average results in the rebuttal version.

---

### Official Review · AnonReviewer3 · 2020-10-30
**Model parallelism for NAS is new but there is no evidence showing it enables a large search space.**

**Rating:** 5
**Confidence:** 4

**Review:**

####Summary:
The paper proposed a binary neural architecture search with consecutive model parallelism to tackle the OOM problem for NAS. The method divides forward/backward phases into several sub-tasks and executes thee same type of the sub-tasks together to reduces idle hardware cycles. This approach effectively improves hardware utilization and saves GPU memory.

####Strengths:
The idea of consecutive model parallel is cute, effectively overlapping the pipeline from two models.

The paper is well written.

####Weakness:
-Binary NAS is not new. The paper is not clear about how much gain is from binary NAS and how much gain is from the consecutive model parallelism.

-Intuitively, model parallelism is used to support larger models and larger search spaces. However, the paper does not provide strong empirical results on larger models constructed using larger supernets. It can be true that larger search spaces make the search and optimization more challenging. But assuming spending more search and training time, the method should be converging to better models.

-The two-cell based search strategy is quite limited. More recent work on layer-wise search space such as TuNAS and EfficientNet yield better results, more particularly, impressive results on ImageNet. Layer-wise search space can be much larger than the cell-based search space. If the paper does not observe significant performance gain via a larger model from a larger search space using model parallelism, it is not very convincing to adopt such a method.

####Detailed feedback:
-The existing results on better hardware utilization and shorter search time are good but not strong enough.

-The reviewer strongly believe the paper can make a bigger impact by enabling a larger search space and demonstrating SoTA performance on more impactful workloads, like ImageNet. A very good baseline to use is EfficientNet, where the model can scale up easily. The goal of this paper should not be targeting reducing search time for toy problems, but aim for improving search quality over larger and more impactful problems.

-There are many related approaches to reduce search time and improve search efficient, such as a latent space search via NAO, or using a surrogate cost function, or using search space pruning via MCTS. There is very few comparisons or explanations why this approach is better.

---

> ### Author Response · Authors · 2020-11-18
> **Thank you for the constructive suggestions and comments.**
>
> **How much gain is from binary NAS and how much gain is from the consecutive model parallelism?**
>
> According to Figure 3, CMP-NASB can run 1.2 faster than MP-NASB in three GPUs. Both two model parallel approaches apply to the same search algorithm (NASB). Even though the search algorithm is the NASB, the two parallelisms are controlled in the same one. So, the whole gain (1.2X) should result from the consecutive model parallelism itself. According to Table 1, NASB is also 1.6 X faster than NASP (6.44 hours/3.92 hours). The result (1.6 X) only indicates that NASB outperforms NASP and does not contribute to the gain of consecutive model parallel.
>
> **Layer-wise search space can be much larger than the cell-based search space.**
>
> EfficientNet uses the small grid search to determine the scaling coefficient for depth, width, and image resolution and fixes the operators in each network layer.  Our search space determines operations in cell structures. Specifically, the search space of normal cells (6-node DAG) is $$C^2_2 8^2 C^3_2 8^2 C^4_2 8^2  C^5_2 8^2=3\times10^9$$, and the one of reduce cells is $$C^2_2 5^2 C^3_2 5^2 C^4_2 5^2  C^5_2 5^2=7\times10^7$$. EfficientNet did not mention the numbers they use in the grid search. Since EfficientNet used grid search, we believe the cell-based search space would not be smaller than the search space of EfficientNet.
>
> **The paper does not provide significant performance gain via a larger model from a larger search space using model parallelism.**
>
> In Table 2, we expected that larger supernets and more diverse cell structures could find better network architectures. We argued that the search time is not enough for the enormous search space. Instead of using EfficientNet search space, we decide to increase supernet complexity by adding more nodes in the cell DAG. We are running the experiment and hope to see the benefits of CMP in large models and large search space.
>
> **Related approaches to reduce search time and improve search efficient**
>
> We will also add NAO and AlphaX (https://arxiv.org/abs/1805.07440) results in Table 1 of the rebuttal version. Specifically, NAO with weight sharing took 7.2 hours to reach a 3.53% test error, and AlphaX took 288 hours to reach a 2.16% test error. Thank you for pointing out other NAS baselines.

---

> > ### Comment · AnonReviewer3 · 2020-11-23
> > **Thanks for the responses.**
> >
> > The reviewer appreciates the authors' great efforts addressing the reviewer's concerns. More particularly, it would be nice to see the additional comparisons with related work such as NAO and AlphaX.
> >
> > The greatest contribution from model parallelism in NAS, is its ability to address OOM problems when using a super network in a oneshot search. That fundamentally limits related work such as TuNAS [1], as you can tell TuNAS can only work for small models. The reviewer looks forward to seeing progresses along that direction.
> >
> >
> > [1] Can weight sharing outperform random architecture search? An investigation with TuNAS, https://arxiv.org/abs/2008.06120

---

> ### Comment · Area_Chair1 · 2020-11-23
> **Feedback necessary**
>
> Dear reviewer,
>
> The authors have responded to your comments below. Could you please go over the response and give feedback to the authors sometime soon? The interactive discussion deadline is this Tuesday and you will not be able to interact with the authors after the date.
>
> Thanks,
> AC

---

### Official Review · AnonReviewer1 · 2020-11-08
**needs stronger evaluation**

**Rating:** 5
**Confidence:** 4

**Review:**

This paper proposes NASB-CMP, which overlaps sub-tasks of forward and backward phases to reduce idle time across GPUs and utilize binary architecture parameters to reduce GPU utilization for heavy supernets. Experiments on CIFAR-10 show NASB- CMP runs 1.2× faster with a large batch size of 896 than other model parallel approaches in 4 GPUs. The large memory usage for NAS algorithms is a critical issue.  The Consecutive model parallel (CMP) overlaps the two forward sub-tasks and two backward sub-tasks is novel, while the path-level binarization is not new. I wish to see a more solid evaluation: the accuracy on the cifar dataset doesn't show advantage over baseline algorithms, and the number of parameters is larger. Although the search cost is significantly reduced, people care about the quality of the search. It's hard to convince people that this approach is better. Given Cifar has a lot of randomness, most NAS algorithms need to demonstrate the effectiveness on ImageNet. The paper needs a more convincing evaluation.

---

> ### Author Response · Authors · 2020-11-18
> **Thank you for the feedback.**
>
> To search networks on ImageNet, we followed the setting of PC-DARTS (https://openreview.net/forum?id=BJlS634tPr) where search epochs are 50, and architectural parameters are frozen in the first 35 epoch (warm-up). Before searching, we randomly sample 10% and 2.5% images from the 1.3M training set of ImageNet as our training set (128,118) and validation set (31,581), respectively. Our search batch size search is 384. Currently, the proposed NASB-CMP only took 7 hours to search networks on four GTX 1080 TI GPUs, while PC-DARTS needs to take 11.5 hours on eight Tesla V100 GPUs. This result demonstrates our search algorithm is also efficient in the large dataset. Now, we are retraining the searched networks from scratch to see its performance. Unfortunately, the retraining phase is much more time-consuming than the search phase. We hope we can finish the retraining before Nov 24th. If the results can finish before the rebuttal deadline, we will add the results to the paper.

---

> ### Comment · Area_Chair1 · 2020-11-23
> **Feedback necessary**
>
> Dear reviewer,
>
> The authors have responded to your comments below. Could you please go over the response and give feedback to the authors sometime soon? The interactive discussion deadline is this Tuesday and you will not be able to interact with the authors after the date.
>
> Thanks,
> AC

---

### Comment · Area_Chair1 · 2020-11-23
**The end of the interactive discussion phase approaching**

Dear Reviewers,

The authors have provided detailed responses to your comments. Could you please go over the responses from the reviewers and provide feedback since the authors can have interactions with you only by this Tuesday (24th)?. I sincerely thank you for your service in reviewing for ICLR.

Thanks,
Area Chair

---

### Decision · Program_Chairs · 2021-01-07
**Final Decision**

**Decision:**

Reject

**Comment:**

This paper proposes a model parallelism scheme (CMP) for training differentiable NAS with large supernets, which performs the forward and backward passes for multiple tasks at the same time, to increase hardware utilization. Moreover, since CMP consumes large GPU memory due to having multiple computational graphs in memory at the same time, the authors further propose binary neural architecture search (NASB), which binarizes the parameters and gradients to reduce memory footprints and computations. The experimental validation shows that the proposed parallelization technique is more efficient than prior parallelization techniques and can significantly reduce the search cost of differentiable NAS methods. Specifically, the proposed NASB-CMP yields architectures with competitive performance on CIFAR-10, at lower search cost compared with baselines that have memory reduction mechanisms such as PC-DARTS.

This paper received split reviews, with the majority of the reviewers leaning toward rejection (three 5’s) and one leaning positive (6). The reviewers in general agreed that the problem of achieving resource and time-efficiency for NAS is important, and that the proposed idea of consecutive model parallelism is novel and may have some practical impact. The reviewers also found the paper to be mostly clear and well-written.

However, reviewers had a common concern that the experimental validation is weak, since 1) the improvement in search cost seems less meaningful with small workloads such as CIFAR-10, 2) some important baselines are missing, and 3) unclear contribution of CMP and NASB due to a missing ablation study. During the interactive discussion period, the authors provided results on additional baselines (NAO and AlphaX), and intermediate results of ImageNet experiments which shows that the proposed method is faster than the baseline. Yet, the reviewers kept their original ratings even after the internal discussion, as they found the incomplete experiments and missing ablation study unsatisfactory.

In summary, this is a well-written paper proposing a novel idea to tackle the resource and time-efficiency of differentiable NAS, which is a practically important problem. Yet, the experimental validation is too weak to validate the effectiveness and practicality of the proposed method, and thus it seems like a preliminary work not yet ready for publication. However, the work is well-motivated and promising, and addressing the reviewers’ common concerns on missing large-scale experiments and ablation study will make the paper stronger and significantly increase its chance of getting accepted in the next submission.